# High-Precision Plane Detection Method for Rock-Mass Point Clouds Based on Supervoxel

**DOI:** 10.3390/s20154209

**Published:** 2020-07-29

**Authors:** Dongbo Yu, Jun Xiao, Ying Wang

**Affiliations:** School of Artificial Intelligence, University of Chinese Academy of Sciences, No. 19 Yuquan Road, Shijingshan District, Beijing 100049, China; yudongbo16@mails.ucas.ac.cn (D.Y.); ywang@ucas.ac.cn (Y.W.)

**Keywords:** plane detection, high-precision, rock mass, voxel, supervoxel, patch-based, region growing

## Abstract

In respect of rock-mass engineering, the detection of planar structures from the rock-mass point clouds plays a crucial role in the construction of a lightweight numerical model, while the establishment of high-quality models relies on the accurate results of surface analysis. However, the existing techniques are barely capable to segment the rock mass thoroughly, which is attributed to the cluttered and unpredictable surface structures of the rock mass. This paper proposes a high-precision plane detection approach for 3D rock-mass point clouds, which is effective in dealing with the complex surface structures, thus achieving a high level of detail in detection. Firstly, the input point cloud is fast segmented to voxels using spatial grids, while the local coplanarity test and the edge information calculation are performed to extract the major segments of planes. Secondly, to preserve as much detail as possible, supervoxel segmentation instead of traditional region growing is conducted to deal with scattered points. Finally, a patch-based region growing strategy applicable to rock mass is developed, while the completed planes are obtained by merging supervoxel patches. In this paper, an artificial icosahedron point cloud and four rock-mass point clouds are applied to validate the performance of the proposed method. As indicated by the experimental results, the proposed method can make high-precision plane detection achievable for rock-mass point clouds while ensuring high recall rate. Furthermore, the results of both qualitative and quantitative analyses evidence the superior performance of our algorithm.

## 1. Introduction

Plane detection plays a significant role in computer vision, the effective detection of planar structures is indispensable for various visual applications, for example, 3D reconstruction [1], object recognition [2], mapping [3], etc. In recent years, with the rapid development of remote sensors, remote sensing technique has been regarded as an effective tool for capturing the high-precision 3D information on targets from a considerable distance, which is conducive to generating the 3D point cloud [4,5]. 3D point cloud data, as a dense collection of points used to depict the surface characteristics of the target, is structureless. Consequently, the extraction of planar structures from 3D laser point clouds has been made a new direction of research on computer vision [6,7,8,9].

As far as rock engineering is concerned, the 3D reconstruction of rock-mass point cloud represents one of the major problems to be solved by computer vision technology. Since the methods like block theory [10] and discontinuous deformation analysis method [11] have been demonstrated as effective in dealing with the analysis of rock-mass numerical models for their stability [12], the role that 3D reconstruction technology plays in rock mass is highlighted. However, it is difficult to obtain high-precision numerical model. As the rock surface is rough and unpredictable, the direct reconstruction of rock-mass point cloud (such as mesh reconstruction) will make the model excessively complex. The excessive information makes it challenging to achieve a precise cutting and analysis of the model. In this case, there is a necessity to compress the original data, which can be achieved by plane detection. The original data can be simplified by detecting and extracting the main structure in the point cloud. The detected plane will be treated as the principal component of the model, thus making it possible to establish a lightweight numerical model.

It is worth noting that, the plane-based approach to model reconstruction places demanding requirements on the results of plane detection. In the course of detection, both less-segmentation and over-segmentation need to be avoided as much as possible. The former will compromise the accuracy of the model, while the latter will cause a heavy burden on the closed work to be performed by the model, since there is still no effective method in achieving the fully automatic model closure. Furthermore, as the ultimate objective of rock-mass reconstruction is to construct a watertight model, the effective detection area in the detection result should cover the original data as much as possible, so as to effectively reduce the manual repair work durning reconstruction. Particularly, it will be a tough challenge to develop a watertight model based on the plane extraction results when the recall rate is low to a certain level. Therefore, it is necessary to maintain a high recall rate when performing plane detection with 3D reconstruction as the target.

In view of the aforementioned factor, an efficient approach to high-precision plane detection from rock-mass point clouds is proposed in this paper. Figure 1 shows an example of our detection. For each input point cloud, specified-resolution spatial grids are adopted for fast voxelization, while high-precision growth units are extracted by means of coplanarity test and accurate edge information calculation. With growth units taken as the seeds required for generating supervoxel, the accurate division of scattered points is made achievable. The final results are obtained by expanding supervoxel patches.

The contributions of this paper are summarized as follows.

An edge optimization method based on iterative detection and similarity measurement is proposed to achieve the accurate identification and processing of edge areas. Meanwhile, part of boundary problems can be effective.A supervoxel segmentation method suitable for rock-mass plane detection is applied. During the process of segmentation, the details of the point cloud are preserved through the accurate judgment of scattered points. Supervoxel segmentation can be performed to obtain high-precision results without over-segmentation. Meanwhile, the supervoxel patches carrying more effective information provide more reliable growth basis for region growing.A patch-based region growing strategy is developed. This method realizes the expansion of the plane from the interior to the edge region, thereby the impact of accumulated error on the merging calculation can be reduced to the minimum level. Meanwhile, the effective adjustment of parameters to the detection results is achieved.

This paper is structured as follows. In Section 2, the related work is introduced. Then, the proposed method is described in detail in Section 3. In Section 4, the experimental results are presented and discussed. In Section 5, a summary is made and the future work is indicated.

## 2. Related Work

Traditional plane detection methods can be divided into two categories, with one based on model fitting and the other premised on neighborhood information. This section mainly introduces the principle of these methods and the relevant improvements made to them. Furthermore, the existing plane detection methods applied to the rock mass are summarized.

### 2.1. Model Fitting-Based Methods

As for the category that is based on model fitting, there are two commonly applied algorithms, including the Hough Transform (HT) [13] and the Random sample consensus (RANSAC) [14].

#### 2.1.1. RANSAC

As a method based on model fitting, RANSAC was first proposed by Fischer and Bolles [14]. When RANSAC is applied to extract a plane, a plane model is generated by selecting three points on a random basis. Then, the score function is applied to determine the matching degree of the remaining point set with this model, while the best plane is detected by iterating the above-mentioned steps. The time complexity of RANSAC is O(I|P|), where *I* represents the number of times for iterations required to detect a plane. Since the random points selected by RANSAC are independent of each other at each time of iteration, the number of iterations is in theory determined by the probability that the best plane can be obtained by an one-off sampling. It is supposed that the whole set of points contains “*N*” points and the maximum plane (the best plane) is comprised of “*n*” points. Let “*w*” denote the probability that each time a point selected from the data set belongs to the best plane, as shown in Equation (Equation 1). Let *P* indicate the probability that one sample can obtain the best plane, as shown in Equation (Equation 2).
(1)$$w=\frac{n}{N}$$
(2)$$P\approx w^{3}={\left(\frac{n}{N}\right)}^{3}$$

It can be found out from above that RANSAC achieves an extremely high efficiency in dealing with simple structures [15]. In practical applications, however, RANSAC is not supposed to be used directly for the data sets with a high degree of discreteness. Progressive Sample Consensus (PROSAC) represents a typical method of improvement as proposed by Chum and Matas [16]. By changing traditional random sampling into semi-random sampling based on similarity, it can reduce the number of samples required while enhancing the efficiency of the algorithm. In order to improve the robustness of RANSAC, Zisserman et al. [17] made RANSAC shift from solving the estimation of maximum likelihood to solving the minimum cost problem. With regard to the detection accuracy of RANSAC as another factor that is concerning, Xu et al. [18] proposed a weighted RANSAC segmentation method, where high-quality detection was performed by setting a weighting function with the best screening performance for abnormal suppression rate. In order to ensure the correctness of sampling on the same plane, Li et al. [19] applied normal distribution transformation cells as the smallest sample of the plane.

#### 2.1.2. Hough Transform

Hough Transform [14] is also categorized as model fitting-based method for plane detection, and is commonly used to detect geometric primitives contained in the point cloud [20]. Currently, a universal version of the Hough Transform for plane detection is the Standard Hough Transform (SHT) [21], which involves two basic steps: Hough voting and peak detection. Each point in the point cloud is converted into a sinusoidal surface in Hough space and voted for the corresponding accumulator. After all the points are voted, the maximum value is extracted from the Hough space and the corresponding geometric parameters are obtained. Though Hough Transform is effective in achieving high-precision detection, its performance is constrained by the astronomical computational cost. The early optimization required by this method mainly focuses on improving the sampling method, including Probabilistic Hough Transform (PHT) [22], Adaptive Probabilistic Hough Transform (APHT) [23], Progressive Probabilistic Hough Transform (PPHT) [24], Randomized Hough Transform (RHT) [25] and so on. Subsequently, in order to effectively reduce the computational cost incurred by Hough Transform, Vosselman et al. [26] divided the split task into the transformation of the normal vector and the transformation of distance. A reasonable simplification of the target point cloud provides another means to improve the performance of Hough Transform. For example, Limberger et al. [27] suggested an octree-based plane detection approach. In their work, PCA was adopted in each local space for the extraction of approximate coplanar samples. Further with that, a kernel-based Hough Transform [28] strategy was put forward to ensure the detection of planar regions in point clouds. Xu et al. [29] proposed a robust plane segmentation algorithm intended to address the photogrammetric point clouds in a construction site scenarios. A 3D Hough Transform-based method, constrained by an orientated bounding box, was developed to construct the initial model of the planes in this method. High precision detection results were obtained by positioning and optimizing the model.

### 2.2. Neighborhood Information-Based Methods

Region growing [30] is one of the most significant methods based on neighborhood information. In addition, as an improved region growing method, supervoxel segmentation proves effective in enhancing the accuracy of segmentation, which makes it widely used in various scene segmentation tasks.

#### 2.2.1. Region Growing

As a classic segmentation method in the field of computer vision, region growing is effective in processing the data with complex characteristics [31]. The selection of seeds and the establishment of growth rules are the two significant influencing factors for region growing. Through a merger of the regions with similar features, the final results are extracted. However, there is a possibility that region growing gives rise to various boundary-related problems [32] and results in precision loss, which is due to the inability of a single seed and fixed growth parameters to effectively constrain the growth process in all directions. To improve the efficiency of region growing, Poppinga et al. [33] suggested the incremental version of the plane fitting formula that can be calculated linearly, with the polygonization step optimized using the neighborhood relationship in the image. Xiao et al. [34] adopted a seed selection method based on local shape information to prevent the blind growth of seeds. Vo et al. [35] proposed a region growing method based on octree segmentation. The major segments were extracted swiftly by means of voxel-based region growing, and then high-precision detection results were obtained by going through the optimization process. With plane fitting as the area growth technique, Wang et al. [36] improved and updated the original algorithm by optimizing the plane calculation and mean square error calculation. Xu et al. [37] developed a pipeline of the accurate plane segmentation for point clouds. According to this method, the whole segmentation process is split into local phase and global phase, based on which the overlapped areas between different planes are detected and segmented accurately with the addition of constraints. Particularly, a new optimal-vector-field is involved to detect the plane intersections in the local phase.

#### 2.2.2. Supervoxel Segmentation

To address the problem caused by traditional region growing, supervoxel segmentation is proposed, which can be regarded as an exceptional solution to region growing. Unlike traditional methods, there are multiple seeds selected from the same plane at the time of supervoxel segmentation and the effective range of each seed is subject to strict restriction. This method is capable to achieve over-segmentation of point clouds, thus preserving the details [38]. Such an idea of point cloud segmentation is of great significance to the method proposed in this paper. As a well-known and effective solution [39], voxel Cloud Connectivity Segmentation (VCCS) allows all points to be evenly distributed to supervoxel patches through similarity measurement-based breadth-first growth strategy. Song et al. [40] applied a supervoxel segmentation method based on boundary enhancement, to detect the boundary by analyzing the continuous points to merge the discrete points. Based on simple hill-climbing optimization, Michael et al. [41] came up with a new approach. The calculation cost incurred by clustering was reduced by defining a fast energy function based on enhancing the color similarity between the boundary and the color histogram of the supervoxel. After selecting the appropriate seeds by calculating the local smoothness of each point, Lin et al. [42] performed K-means clustering to normalize the boundary of supervoxel patches.

### 2.3. Methods For Rock-Mass Point Clouds

In recent years, some effective plane detection methods intended for rock-mass point clouds have been proposed, for example, Riquelme et al. [43] developed the Discontinuity Set Extractor (DSE) software. Meanwhile, a semi-automatic rock mass discontinuity extraction method was adopted. It relied on PCA to identify the coplanarity of neighboring points, with high-precision detection results obtained using HT. Leng et al. [44] proposed a multi-scale rock surface detection method based on HT and RG, with multi-scale plane detection performed by adjusting parameters manually. Liu et al. [45] put forward an efficient detection strategy, where coplanarity test was conducted to simplify point clouds in an efficient way. Then, HT was used exclusively to calculate the major orientation in coplanar patches, while the surfaces of rock mass were extracted by hybrid region growing (HRG). Based on RANSAC and RG, a similar approach was also presented by Hu et al. [46]. Wang et al. [47] suggested an effective method to extract rock fractures using regional growth. According to this method, the criteria based on the local surface normal and curvature of the point cloud are applied to initiate and control the growth of the fracture region. In order to evaluate the discontinuity geometric properties from a point cloud, Ge et al. [48] applied a modified region growing (MRG) algorithm. The MRG is characterized by the grow criterion with a higher efficiency. Vasuki et al. [49] developed an interactive image segmentation algorithm. In accordance with this method, the user is expected to draw rough markings to indicate the locations of different geological units in the image. Image segmentation is performed by a number of segmenting regions based on their homogeneity in color. Based on a 3D surface model of rock mass, Li et al. [50] presented an automated discontinuity trace mapping method, where the feature points of discontinuity traces were first detected using the Normal Tensor Voting Theory, before the extraction of discontinuity traces from feature points. Jang et al. [51] used color and normal vector of the point clouds to distinguish surfaces and edges of rock, which realized the accurate measurement of rock fragmentation.

Compare with the methods introduced in this section, our approach is more advantageous in achieving high precision and high recall rate plane detection while ensuring efficiency. In addition, stable and adjustable results can be achieved through the patch-based region growing.

## 3. Methodology

The method proposed in this paper consists of three main parts, as shown in Figure 1. According to this method, the major segments of planes are extracted by means of coplanarity test and edge information calculation based on voxels, with the plane detected in each voxel space treated as an independent growth unit. Then, supervoxel segmentation is adopted to effectively deal with the remaining scattered points, with each growth unit viewed as a seed. The completed planes are obtained by merging similar supervoxel patches through the patch-based region growing.

### 3.1. Growth Units Extraction

In the first part of the method, RANSAC-based coplanarity test is conducted after voxelization to extract high-precision growth units, with these growth units comprising the major segment of the plane. With a search of clear planar structures in each local space, the number of unorganized points can be reduced significantly, thus improving the efficiency of the algorithm.

#### 3.1.1. Coplanarity Test Based on Voxels

At the start of the algorithm, the spatial grids with specified-resolution are employed to perform the voxelization of point clouds, which are purposed to divide the point cloud with a complex structure into multiple simple subareas, thus significantly enhancing the performance of local calculation. To restrict the number of points in each voxel space, it is necessary to take into account the density and scale of point cloud when setting a resolution. Meanwhile, the neighborhood index is established for each voxel according to the positional relationship between spatial grids.

Principal Component Analysis (PCA) is a classic method for coplanarity test, but with its performance constrained by the precision of voxelization. In addition, due to the inability to deal with the boundary effectively, PCA is incapable to conduct a thorough analysis of the object. In our study, the coplanarity test was conducted by extracting planes in each subspace, so as to ensure the effective detection of non-coplanar regions. RANSAC was chosen for local calculation due to its clearly superior performance in processing subspace with simple structures.

RANSAC-based coplanarity test was carried out once in each voxel space, while each extracted plane will be treated as an independent growth unit. All points contained in a growth unit will be considered as a whole to perform the subsequent calculation, thus improving the efficiency of the detection. It is noteworthy that, in order to obtain a high-precision growth unit, it is essential to set the angle threshold and distance threshold with high-precision should in the course of test. The details of this part are shown in Algorithm 1.
**Algorithm 1** Coplanarity test based on voxels.**Input:** {P}: input point cloud, $Voxel_{size}:$ size of space grid, $Min_{plane}:$ the minimum number of points in a plane;**Output:** {Grow_unit}: set of growth units, {Point_remain}: set of remaining points after RANSAC;1:{voxel}← segmenting {P} according to $Voxel_{size}$;2:**for** each $voxel_{i}$ of {voxel} **do**3:    $grow\_unit_{i}\leftarrow$ the best plane in the $voxel_{i}$.{extracted by RANSAC};4:    $point\_remain_{i}\leftarrow$ remaining points in the $voxel_{i}$;5:    **if**
$|grow\_unit_{i}|<Min_{plane}$
**then**;6:        insert $grow\_unit_{i}$ into $point\_remain_{i}$;7:        add $point\_remain_{i}$ into {Point_remain};8:    **else**9:        add $grow\_unit_{i}$ into {Grow_unit};10:        add $point\_remain_{i}$ into {Point_remain};11:    **end if**12:**end for**

#### 3.1.2. Edge Information Calculation

When a subspace contains a minimum of two planes, the above-mentioned steps cannot deal with all unorganized points effectively. In rock-mass point clouds, such non-coplanar voxels are concentrated near the boundary between planes, as shown in Figure 2, which reveals that each non-coplanar voxel contains the edge regions of multiple planes. Figure 3a lists the results of the coplanarity test for non-coplanar voxels. The best plane (orange plane) in the non-coplanar voxel can be extracted, while the edge regions of other planes remain in the form of discrete points. When rock-mass point clouds are dealt with, it is possible that such a phenomenon cause boundary issues, as shown in Figure 3b, which is because a greater degree of discrepancy between discrete points and the target plane is allowed for the subsequent processing. Therefore, a further detection of non-coplanar voxels is deemed necessary.

The edge coplanarity test based on RANSAC is repeatedly performed in non-coplanar voxels, for which an effective test is required to satisfy two conditions as follows: (1) There are a sufficient number of remaining points in a voxel; (2) The detected plane contains more points than the threshold. With the above-mentioned two conditions imposed, most non-coplanar voxels can perform the effective test only once, with the result shown in Figure 3c.

The edge patches as obtained from iterative coplanarity test are not the best plane in the subspace. To improve the robustness of our algorithm, the edge patches with fewer points need to be incorporated into those connected growth units instead of being treated as an independent growth unit. It is worth mentioning that the neighborhood calculation involving boundaries must be treated with caution, which is because a critical step in segmenting planes is to determine the edge patch. Therefore, the angle threshold and distance threshold with high-precision are applied to approach the determination. Additionally, the judgment method based on similarity measurement is adopted to preserve all potential boundaries in an accurate way. Since the essence of patches and growth units is small plane, the distance and the angle between patches and growth units can be calculated using Equations (Equation 3) and (Equation 4), respectively.
(3)$$m_{w}=\frac{1}{n}\sum_{i=1}^{n}\frac{\left|Ax_{i}+By_{i}+Cz_{i}^{}+D\right|}{\sqrt{A^{2}+B^{2}+C^{2}}}$$
(4)$$n_{w}=f\left(n_{1,}n_{2}\right)=\arccos\left|\frac{n_{1}n_{2}}{\left|n_{1}\right|\left|n_{2}\right|}\right|$$
where $n_{1}$ and $n_{2}$ represent the normal vectors of two planes, *A*–*D* indicate the parameters of one of the plane, $\left(x_{i},y_{i},z_{i}\right)$ denote coordinates of the points in the other plane. If there are more than one growth units satisfying the merging conditions (the actual values are smaller than thresholds), using similarity measure is token to determine the assignment of patches. The similarity $P_{w}$ of two planes can be calculated using Equation (Equation 5), where $M_{w}$ and $N_{w}$ are the thresholds.
(5)$$P_{w}=\sqrt{{\left(\frac{m_{w}}{M_{w}}\right)}^{2}+{\left(\frac{n_{w}}{N_{w}}\right)}^{2}}$$

The equation measures the differences between two planes in the dimensions of distance and angle. The smaller the value of $P_{w}$, the higher the degree of similarity, so as to provide a reliable basis for the precise merging of edge patches. The specific process of edge information calculation is detailed in Algorithm 2.
**Algorithm 2** Edge information calculation.**Input:** {Grow_unit}, {Point_remain},$M_{w}$,$N_{w}$,$Min_{edge}$,$Ran_{edge}$;**Output:** {Grow_unit}, {Point_remain};1:**for** each $point\_remain_{i}$ of {Point_remain}
**do**2:    $test_{able}\leftarrow 1$;3:    **while**
$|point\_remain_{i}|>Min_{edge}$**and**$test_{able}=1$
**do**4:        $edge\_patch_{i}\leftarrow$ the edge patch extracted by RANSAC;5:        $edge\_remain_{i}\leftarrow$ remaining points after RANSAC;6:        **if** ($|edge\_patch_{i}|>Ran_{edge}$) **then**7:           {Neighbor}← neighbors of $voxel_{i}$;8:           Sim←100;9:           **for** each $Neighbor_{j}$ of {Neighbor}
**do**10:               $grow\_unit_{j}\leftarrow$ the best plane in $neighbor_{j}$;11:               $m_{w}\leftarrow$ distance between $edge\_patch_{i}$ and $grow\_unit_{j}$ { by Equation (Equation 3)};12:               $n_{w}\leftarrow$ angle between $edge\_patch_{i}$ and $grow\_unit_{j}$ {by Equation (Equation 4)};13:               **if**
$m_{w}<M_{w}$**and**$n_{w}<N_{w}$
**then**14:                   $P_{w}\leftarrow$ similarity between $edge\_patch_{i}$ and $neighbor_{j}${ by Equation (Equation 5)};15:                   **if**
$P_{w}<Sim$
**then**16:                       record←j,$Sim\leftarrow P_{w}$;17:                   **end if**18:               **end if**19:           **end for**20:           **if** Sim <100 **then**21:               insert $edge\_patch_{i}$ into $grow\_unit_{record}$;22:               update {Grow_unit};23:           **else**24:               add $edge\_patch_{i}$ into {Grow_unit};25:           **end if**26:           $point\_remain_{i}\leftarrow edge\_remain_{i}$;27:           update {Point_remain};28:        **else**29:           insert $edge\_patch_{i}$ into $point\_remain_{i}$;30:           update {Point_remain};31:           $test_{able}\leftarrow 0$;32:        **end if**33:    **end while**34:**end for**

The final result of edge information calculation is shown in Figure 3d. Due to the accurate processing of edge patches, the junction between different planes will be restricted to a confined area, thus preventing various boundary-related problems. The similarity calculation method proposed in this paper is applied by simplifying VCCS. In particular, the application scope of similarity measure is limited to the judgment of edge patches. Meanwhile, the measured object is shifted from a point to a patch for the purpose of pinpointing the boundaries. The improvement can not only satisfy the requirements for plane detection, but also ensure an effective reduction to the computational cost.

### 3.2. Supervoxel Segmentation

After extracting growing units, the input point cloud is segmented into several growing units (small planes) and a large number of scattered points. Hybrid region growing has been proved to be effective in merging growth units and discrete points with similar types to generate a connected plane. However, since the accuracy of growth in all directions can hardly be effectively controlled by a single seed and constant growth parameters, less-segmentation and boundary problems may occur during segmenting with HRG. The update method of the parameters in processing and the characteristics of the rock mass are the fundamental reasons for uncontrolled growth in some directions. More specifically, the parameters of seeds are obtained by plane fitting using Mean Squared Error (MSE), which can be transformed into calculating eigenvalues and eigenvectors of the corresponding covariance matrix. The covariance matrix *C* can be assembled by Equation (Equation 6). After the extraction of growing units, the input point cloud is split into multiple growing units (small planes) and a large number of scattered points. Hybrid region growing has been demonstrated as effective in merging growth units and discrete points with the similar types to generate a connected plane. Since the accuracy of growth in all directions is basically unlikely to be controlled by a single seed and constant growth parameters, however, it is possible for less-segmentation and boundary problems to arise from the detection with HRG. The method of update to the parameters in processing and the characteristics of the rock mass are the leading causes for uncontrolled growth to occur in some directions. More specifically, the parameters of seeds are obtained by means of plane fitting based on Mean Squared Error (MSE), which can be converted into calculating eigenvalues and eigenvectors of the corresponding covariance matrix. The covariance matrix *C* can be assembled using Equation (Equation 6).
(6)$$C=\frac{1}{k_{o}+k_{a}}\sum_{i=1}^{k_{o}+k_{a}}\left(p_{i}-m\right){\left(p_{i}-m\right)}^{T}$$
where $k_{o}$ represents the number of points in the initial seed and $k_{a}$ to the number of points added into the seed during segmentation, “*m*” indicates the centroid of the plane which can be calculated using Equation (Equation 7).
(7)$$m=\frac{m_{o}k_{o}+m_{a}k_{a}}{k_{o}+k_{a}}$$

The parameters of the plane will be largely determined by the direction of $k_{a}$ when $k_{a}$ is significantly greater than $k_{o}$. Allowing for the size of rock-mass planes, such a phenomenon may occur on a frequent basis. In addition, a certain range of error should be accommodated at each step during detection since the surfaces of rock mass are rugged and unpredictable, as shown in Table 2. Consequently, with the increase of $k_{a}$, such errors will be accumulated rapidly, especially for the merger of scattered points. The initial seed and growth parameters may be rendered useless when the errors are accumulated to a certain degree.

In this paper, a supervoxel segmentation method is designed to solve the above-mentioned problem. The core idea of the method is to break down the region growing into several subtasks for a precise control over the accumulated error. The seed selection method is quite simple, for it is reliable and reasonable to apply the best plane (growth unit) as a seed in each voxel. Furthermore, as the objects to be processed for a seed are restricted to the remaining discrete points in the local space and neighborhoods, the effectiveness of seeds can be guaranteed at all times. To improve the accuracy of supervoxel segmentation and prevent over-segmentation, the cumulative error is supposed to be strictly limited, as shown in Figure 4. In this case, an iterative approach is adopted. At each time of growth, it is supposed that “$f_{1}$” indicates the initial normal vector of the seed and “$f_{2}$” denotes the new normal vector of seed after growing, based on which the angle “θ” between “$f_{1}$” and “$f_{2}$” can be obtained using Equation (Equation 8).
(8)$$\theta =f(f_{1,}f_{2})$$

Different from Equation (Equation 4), Equation (Equation 8) calculates the differences of a single patch before and after growing. With the initial parameters of the seed as the true value, θ can be regarded as the cumulative error, which is caused by merging some scattered points. Only if the accumulated error is less than the threshold can the growing be considered as valid. Notably, the cumulative error is calculated only when a seed stops growing and each valid θ will be recorded, for use in the subsequent steps. If the growth is found invalid, it is necessary to restore all the information involved in this growth. In the meantime, the growth threshold will be updated depending on the set step size for an improved growth precision. The aforementioned process will be repeated until an effective growth is completed. Dissimilar from the real-time monitoring of accumulated errors, the method proposed in this paper can produce the optimal segmentation results within the allowable range of errors by updating parameters iteratively. How supervoxel segmentation proceeds is shown in Algorithm 3.

The supervoxel segmentation method proposed in this paper is accurate in merging the remaining scattered points with the supervoxel patches, as a result of which the details in point clouds can be effectively preserved. In addition, the application of supervoxel patches with more details as growth units are conducive to performing region growing in a more effective way.
**Algorithm 3** Supervoxel segmentation.**Input:** {Grow_unit}, {Point_remain},$super_{dis}$,$super_{ang},Max_{error}$;**Output:** {Supervoxel_patch}, {Error}: the set of errors contained in each supervoxel patch;1:**for** each $grow\_unit_{i}$ of {Grow_unit} **do**2:    $grow\_unit\_record_{i}\leftarrow grow\_unit_{i}$;3:    {Neighbor}← neighbors of $voxel_{i}$ ; 4:    **for** each $Neighbor_{z}$ of {Neighbor}
**do**5:        $remain\_point\_point_{z}\leftarrow$ the remain points in $Neighbor_{z}$;6:        $remain\_point\_record_{z}\leftarrow remain\_point\_point_{z}$;7:    **end for**8:    $super_{i}\leftarrow super_{dis}$ , $super_{i}\leftarrow super_{ang}$ , $\theta_{i}\leftarrow 100$;9:    **while** ($\theta_{i}>Max_{error}$) **do**10:        $grow\_unit_{i}\leftarrow grow\_unit\_record_{i}$;11:        **for** each $Neighbor_{x}$ of {Neighbor}
**do**12:           $remain\_point_{x}\leftarrow$ the remain points in $neighbor_{x}$;13:           $remain\_point_{x}\leftarrow remain\_point\_record_{x}$;14:        **end for**15:        **for** each $Neighbor_{j}$ of {Neighbor}
**do**16:           point_remain← the remain points in $Neighbor_{j}$;17:           **for** each $p_{k}$ of $point\_remain_{j}$
**do**18:               $d_{p}\leftarrow$ distance between $p_{k}$ and $grow\_unit_{i}$;19:               $n_{p}\leftarrow$ angle between $p_{k}$ and $grow\_unit_{i}$;20:               **if**
$d_{p}<super_{i}$**and**$n_{p}<super_{i}$
**then**21:                   insert $p_{k}$ into $grow\_unit_{i}$;22:                   remove $p_{k}$ from $point\_remain_{j}$;23:               **end if**24:           **end for**25:        **end for**26:        $\theta_{i}\leftarrow$ angle between $grow\_unit_{i}$ and $grow\_unit\_record_{i}$;27:        $super_{i}\leftarrow super_{i}-dis_{step}$ , $super_{i}\leftarrow super_{i}-ang_{step}$;28:    **end while**29:    add $grow\_unit_{i}$ into {Supervoxel_patch};30:    add $\theta_{i}$ into {Error};31:**end for**

### 3.3. Patch-Based Region Growing

In our method, the final results can be fast obtained by merging supervoxel patches through patch-based region growing. Assume that “*N*” represents the set of all supervoxel patches boundaries and "*M*" represents the set of plane boundaries in the final detection result. Then the relationship between *M* and *N* can be expressed as Equation (Equation 9), which means no new boundary will be generated during patch-based region growing.
(9)M⊆N

Compared with the hybrid region growing (based on both points and patches), the patch-based region growing can obtain more stable output, for fewer parameters should be adjusted. Moreover, the detection results with different precision can be extracted by simple parameter adjustment (as shown in Figure 5).

The impact of errors on region growing has already been discussed in Section 3.2. Therefore, a further consideration given to errors is necessary. In our study, the cumulative error is of practical significance, since the accumulation of errors can be effective in reflecting the trend of changes to the surface structure in the region, thus providing a reliable basis for not only seeds selection but also the development of growth rules. In order to determine the order of processing for patches, all the patches are sorted in ascending order based on the value of θ. During each growth, the first unmarked patch in the queue will be taken as a seed, as a result of which region growing can start in the region with a relatively consistent planarity. The extension of seed will start from the neighborhood and the patch will be marked when merged with the seed. Meanwhile, the neighborhood set will be updated and only unmarked patches will be considered at the time of updating. When new patches are added to the set, the order of patches in the neighborhood set will be reset depending on the size of θ, which makes the order of processing for the patches consistent with the original setting. The process of region growing is detailed in Algorithm 4.

In view of the limitations imposed on a single seed for determining the range of a rock-mass plane, only the initial position of seeds is constrained in the proposed method, as shown in Figure 6. In addition, since the order of the patches to be tested is constantly updated, the growth process does not revolve around seeds. Instead, it preferentially spreads to the regions with a stable structure. This method is effective in exercising control on the cumulative speed of errors in the growth process, thus eliminating the impact of errors on parameter calculation to the greatest possible extent. In addition, the growth results in the interior of the plane will provide a reliable basis for determining edge patches, which makes the detection results of complex regions (as shown in Figure 5) consistent with the parameters.
**Algorithm 4** Patch-based region growing.**Input:** {Super_voxel_patch}, {Error};**Output:** {Plane};1:{Super_voxel}← sort {Super_voxel} in ascending order according to the size of θ{based on {Error}};2:**while**{Super_voxel} is not empty **do**3:    seed← the first supervoxel patch in {Super_voxel};4:    {Neighbor}← the neighbor supervoxel patches of seed;5:    **while**
{Neighbor} is not empty **do**6:        {Neighbor}← sort {Neighbor} in ascending order according to the size of θ;7:        patch← the first supervoxel patch in {Neighbor};8:        dis← distance between patch and seed;9:        ang← angle between patch and seed;10:        **if**
$dis<dis_{grow}$ and $ang<dis_{ang}$
**then**11:           insert patch into seed;12:           update {Neighbor};13:           remove patch from {Super_voxel};14:        **else**15:           remove patch from {Neighbor};16:        **end if**17:    **end while**18:    add seed into {Plane};19:**end while**

## 4. Results and Discussion

Our method has been implemented in C++, with both qualitative analysis and quantitative analysis conducted to verify the performance produced by the method. In this section, one artificial point cloud and four rock-mass point clouds were taken as the experimental data. In addition, seven different plane detection methods were applied for contrast experiments, including three traditional methods and four methods for the rock mass. All experiments were performed on Intel (R) core i3-6100 3.60 GHz CPU and 4.00 GB ram in the absence of any parallel operation.

### 4.1. Dataset

For an accurate evaluation of the algorithm for its performance, an icosahedron point cloud and four rock point clouds were taken as the experimental data in this paper, as shown in Figure 7. The icosahedron point cloud was artificially generated, which is aimed to measure the effect of this method in artificial scenes and verify the correctness of the code. Four rock-mass point clouds were all obtained from the Rockbench repository [52], using a Leica HDS6000 scanner in Kingston, Canada. Table 1 shows the basic information on all of the point clouds.

### 4.2. Parameter Analysis

The premise for the algorithm to operate normally is to set reasonable parameters. According to this method, the main parameters involved include the threshold value of the judgment between points and patches, and between patches. The main parameters used in this paper and the relevant instructions are presented in Table 2, with the main ideas of the method sorted out. The algorithm starts with local computation, enriches local information through neighborhood computation and ends up performing global computation. The precision of the parameters shows a downward trend, thus ensuring that the points with different degrees of dispersion can be detected by appropriate parameters. Especially, compared with the existing method, high-precision parameters can be used for the local computation performed using this method due to the addition of neighborhood calculation, which contributes significantly to performance improvement.

### 4.3. Related Methods

In this paper, there are seven methods inolved in comparative experiments, which can be divided into two categories. RG [33], RHT [53] and RAN-PCL [54], respectively represent three traditional plane detection methods. HT-RG [44], DSE [43], MOE [45], RAN-RG [46] are four plane detection methods for rock mass. The basic principle of each method can be found in Section 2.3.

### 4.4. Artificial Point Cloud Detection

To validate the method proposed for artificial scenes and verify the correctness of the code, an artificial icosahedron point cloud was applied for comparative experiments. Figure 8 lists the results of the icosahedron processed by the proposed method and seven comparison methods, while the amounts of time required for all methods are shown in Table 3, which reveals that all the eight methods are effective in performing the plane detection of the icosahedron. As the surfaces of the icosahedron are flat and free of any noise, the difference of detection results mostly exists in the boundary. Compared with other methods, the detection results of our algorithm contain more accurate and regular boundaries. In terms of efficiency, the length of time required for other methods is shown to be acceptable, except DSE and HT-RG.

### 4.5. Rock-Mass Point Clouds Detection

In this section, four rock-mass point clouds are employed for evaluating the proposed algorithm. To verify the algorithm for its performance, both qualitative and quantitative analysis were adopted for the Rock1 detection result. Meanwhile, the detection results of the other three rock-mass point clouds were obtained.

#### 4.5.1. Qualitative Analysis

The analysis of surface structure is conducive to evaluating the performance of the algorithm in an accurate way. Therefore, Rock1 was manually segmented in the first place through visual observation to generate reference point cloud, as shown in Figure 9a. It is worth noting that the reference surface may not be completely accurate, which is because the consistency of scale can barely be guaranteed in artificial segmentation. In spite of this, the results of artificial segmentation can reflect the surface structure of most regions faithfully. In the meantime, a possible segmentation scheme was proposed in the disputed region. Rock1 shows typical rock-mass characteristics, with the number of points and the span of Rock1 shown in Table 1. In addition, it contains not only large span planes, but also some small planes (plane1). Meanwhile, some planes are contained in the black region, since they are too small to be judged intuitively. The plane relation in Rock1 is relatively complex. In addition to a large number of clear joint surfaces, it contains several sets of transition surfaces with similar planarity as well (plane1 and plane2, plane3 and plane4, plane5 and plane6). It is worth mentioning that the detection results of these planes are uncertain at the time of processing, and it depends on the parameters whether they are divided or merged. Therefore, the reference point cloud shown in Figure 9a can represent the result of a high precision detection, which can be used to validate the algorithm when high precision results are needed.

Figure 9 shows the detection results of Rock1 using different methods. Apparently, traditional methods are subject to significant limitations on the segmentation of rock-mass point clouds. It is difficult for RG to deal with the boundary effectively. RHT has a possibility to cause a lot of over-segmentation issues, while RAN-PCL will detect some planes without connectivity. Compared with the traditional plane detection method, the plane detection algorithms intended for rock mass could produce a better performance in dealing with Rock1. To compare these algorithms for their performance, the detection results of plane1 are supposed to be particularly observed. HT-RG and DSE are effective in obtaining the details from the point cloud. Meanwhile, however, they may lead to over-segmentation and missing planes. Though MOE and RAN-RG are accurate in extracting relatively clear joint surfaces, it is difficult for both to segment the regions with similar planarity effectively. Compared with the methods as mentioned above, the proposed method can not only avoid over-segmentation but also preserve details. Moreover, our method is capable to extract very small planes (the black regions in Figure 9), which are not distinguishable for naked eyes, therby the precision and recall rate of detection are further improved.

In order to verify the performance of the algorithm, another three natural rock-mass point clouds were selected for comparative experiments. The rectangular areas shown in Figure 10 evidence the capability of this method to segment continuous patches in an accurate way. To facilitate observation and comparison of the results, the regions that should be observed were marked by ellipses. According to the results, the proposed method is effective in dealing with various complex plane relations, with high-precision results made obtainable without less-segmentation and boundary issues. In addition, as the accumulated error is effectively brought under control, both missing planes and over-segmentation can be avoided.

#### 4.5.2. Quantitative Analysis

To quantify the difference between the results and the reference point cloud, a classic evaluation system is adopted, involving three indicators, accuracy, recall and F1. This evaluation method is widely used for quantitative calculation for its proven effectiveness in evaluating the performance of a system. It was assumed that *a* is a detected plane and *m* is the corresponding reference plane, the true positive (TP) = a ∩ m, the false positive (FP) = a−m and the false negative (FN) = m−a. The above-mentioned evaluation metrics can be calculated using Equations (Equation 10)–(Equation 12).
(10)$$precision=\frac{\left|TP\right|}{\left|TP\right|+\left|FP\right|}$$
(11)$$recall=\frac{\left|TP\right|}{\left|TP\right|+\left|FN\right|}$$
(12)$$F1=2\times \frac{precision\times recall}{precision+recall}$$

With the point cloud in Figure 9a as the reference point cloud, the indicators of different methods were calculated, as shown in Table 4. Additionally, in order to compare the capability of different methods to deal with complex details, the indicator calculation results of plane1 are also listed in Table 5. Though HT-RG and DSE are effective in dealing with details, the performance of these algorithms is subject to impact from over-segmentation. When there is similar planarity between planes, MOE and RAN-RG are incapable to segment such area accurately, while the effectiveness of the above-mentioned algorithm is mainly reflected in the treatment of clear joint surfaces. The method proposed in this paper is capable to achieve high-precision plane detection while ensuring accuracy and recall. The algorithm can produce an evidently superior performance.

Table 6 shows the length of time required by various methods for all rocks, where the high computation cost required by HT-RG and DSE is clearly reflected. When rock-mass point clouds are dealt with, parameter adjustment is necessary, which means consideration shall be given to the effectiveness of parameter adjustment when algorithm efficiency is compared. MOE and RAN-RG can achieve a high detection efficiency. However, it is difficult for both of them to make effective adjustment to parameters, since there are too many parameters involved in HRG. The computation cost incurred by applying our method is acceptable. Meanwhile, the effectiveness of adjustment to each parameter can be ensured by patch-based region growing, as a result of which the efficiency of obtaining ideal results can be ensured.

## 5. Conclusions

This paper focuses on the high-precision plane detection of rock-mass point clouds, where the accurate and thorough segmentation of rock surface is achieved. The core idea of our method is to segment the rock mass into a number of sub-regions with a simple structure. With different methods and parameters used, the points with different degrees of discretization were extracted. Among them, supervoxel segmentation was adopted to deal with the scattered points failing the coplanarity test, as a result of which more effective details were preserved. Furthermore, the stable and adjustable results were obtained by patch-based region growing.

To evaluate the performance of our algorithm, 5 experimental datas and 7 related methods were selected for comparative experiments. In general, the proposed method demonstrated obvious advantages in detection precision and recall rate, as verified by both qualitative analysis and quantitative analysis. In addition, the performance of this algorithm was further improved due to the boundary problem and over-segmentation being solved effectively. Furthermore, the detection results of this algorithm can be adjusted flexibly, thus making the method even more applicable in practice.

The generation of a watertight 3D model is the ultimate goal set for the three-dimensional reconstruction of rock mass, which depends on the analytical results of the surface structure. However, the proposed method can not deal with all the surface structures in some cases. Since the surfaces of rock masses are not all comprised of planes, some structures may be lost when our method for detection is applied. In the future, the results of existing plane detection will be used for 3D reconstruction and the effective repair of the missing part of the surface during the process of reconstruction.

## Figures and Tables

**Figure 1 sensors-20-04209-f001:**
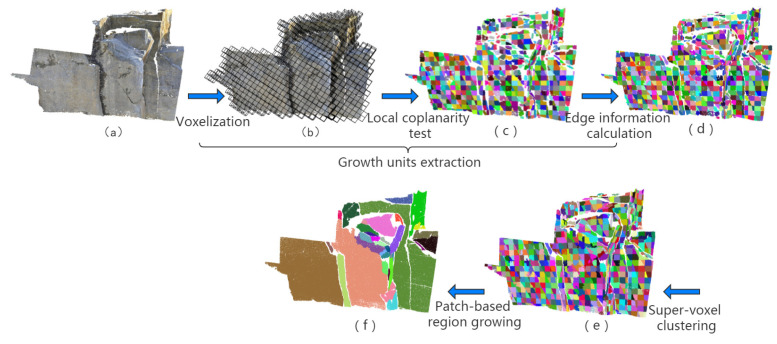
Algorithm flowchart. (**a**) Input point cloud. (**b**) Voxel segmentation. (**c**) Major segment extraction. (**d**) Edge optimization. (**e**) Detail preservation. (**f**) Final results.

**Figure 2 sensors-20-04209-f002:**
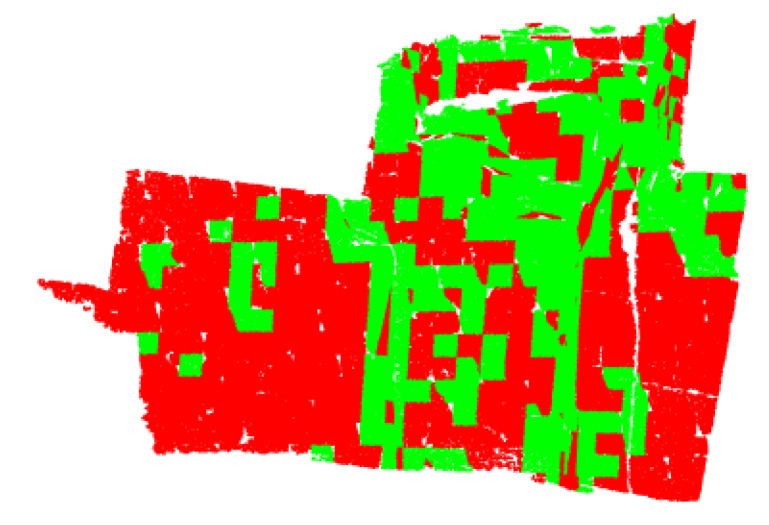
The distribution of non-coplanar voxels(marked in green) in the point cloud.

**Figure 3 sensors-20-04209-f003:**
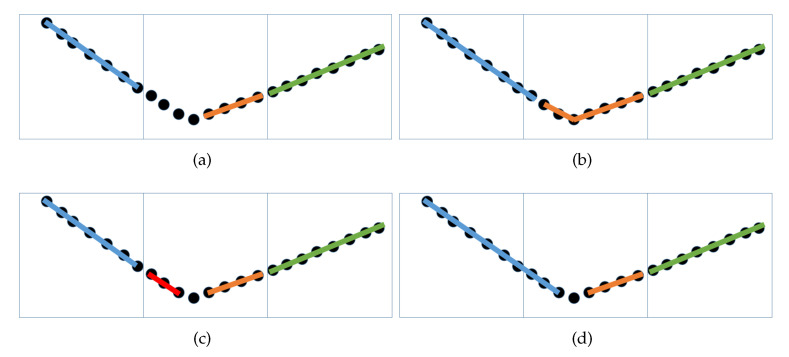
Edge information calculation. (**a**) Result after the coplanarity test. (**b**) An example of boundary problem. (**c**) Edge patch extraction. (**d**) Merging edge patch.

**Figure 4 sensors-20-04209-f004:**
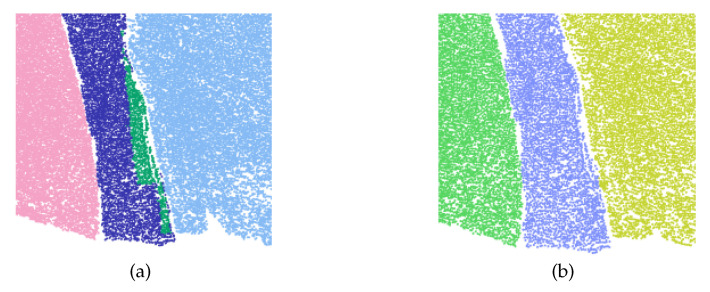
Controlling cumulative errors. (**a**) No limitation; (**b**) θ = $5^{\circ }$.

**Figure 5 sensors-20-04209-f005:**
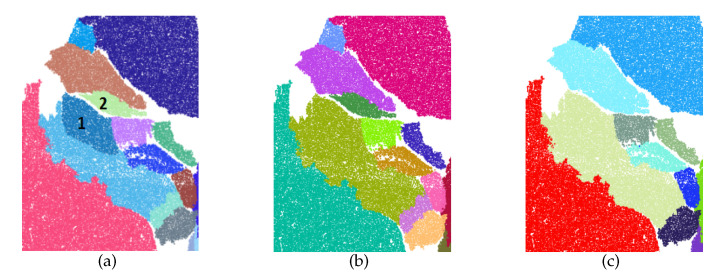
Different precision results obtained by adjusting parameters, where patch1 and patch2 have similar planarity with their neighbors. (**a**) $grow_{ang}=20^{\circ }$; (**b**) $grow_{ang}=22^{\circ }$; (**c**) $grow_{ang}=23^{\circ }$.

**Figure 6 sensors-20-04209-f006:**
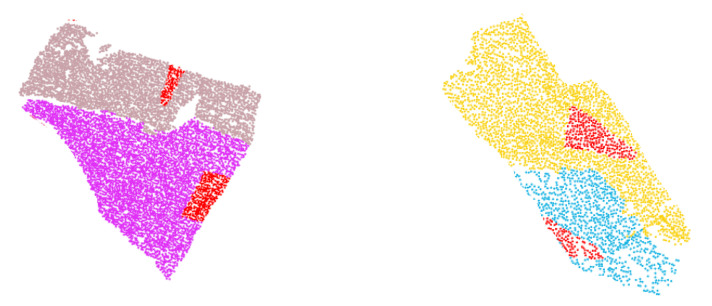
Results of seed selection (Seeds are marked in red).

**Figure 7 sensors-20-04209-f007:**
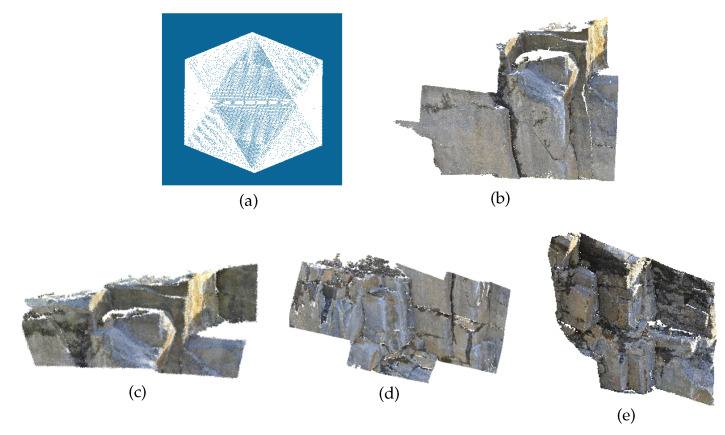
Five datasets. (**a**) Icosahedron; (**b**) Rock1; (**c**) Rock2; (**d**) Rock3; (**e**) Rock4.

**Figure 8 sensors-20-04209-f008:**
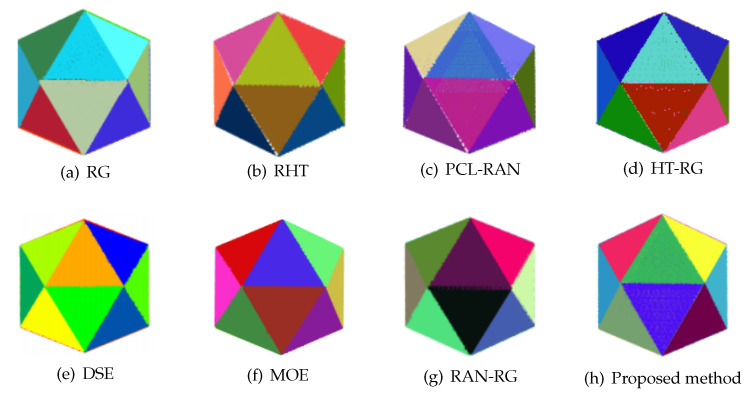
Detection results for the icosahedron.

**Figure 9 sensors-20-04209-f009:**
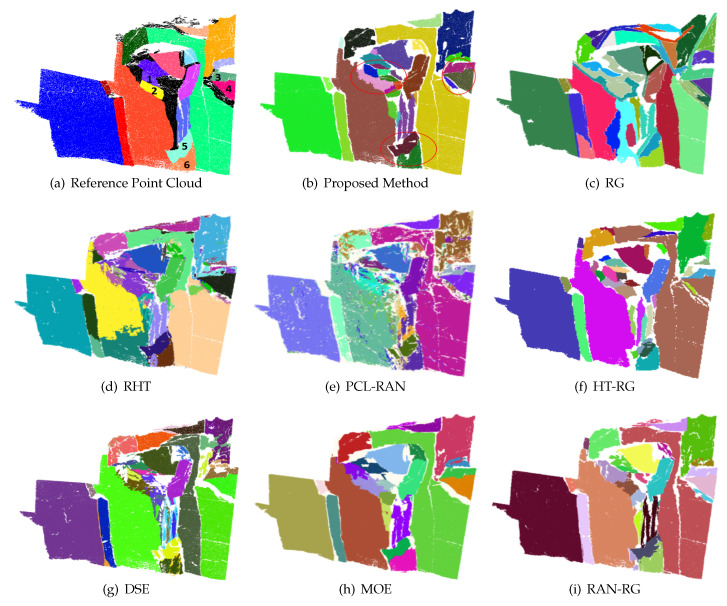
Detection result of Rock1.

**Figure 10 sensors-20-04209-f010:**
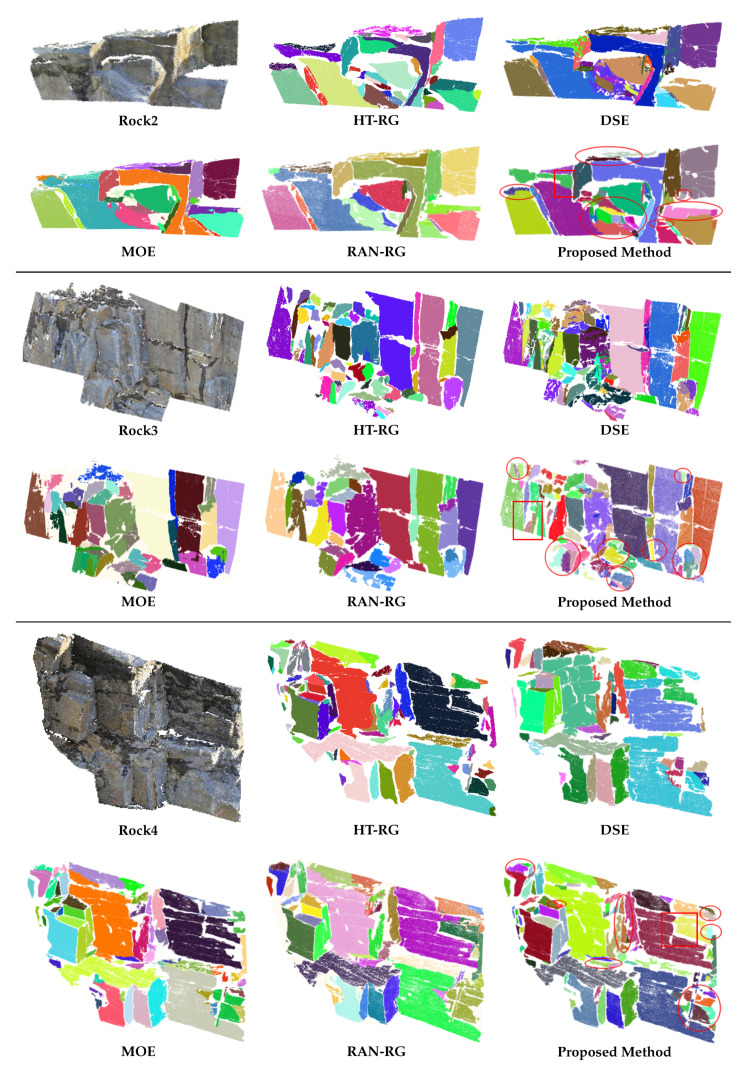
Detection Results of Rock2, Rock3 and Rock4.

**Table 1 sensors-20-04209-t001:** Basic information of point clouds.

Data	Size	Maximum Point Spacing (m)	Minimum Point Spacing (m)	Bounding Box (m)
Icosahedron	92,520	$2.2\times 10^{-16}$	0.243	40 × 40 × 40
Rock1	307,300	0.494	0.029	31.5 × 22.9 × 11.7
Rock2	312,659	0.473	0.019	28.8 × 13.1 × 11.9
Rock3	264,309	0.499	0.049	51.9 × 12.9 × 12.3
Rock4	460,157	0.531	0.031	30.1 × 29.8 × 9.4

**Table 2 sensors-20-04209-t002:** Information of parameters.

Stage	Parameters	Data Types Involved	Effective Range	Values Range
Coplanarity test	$ran_{dis}$	points	local	0.04–0.06 (m)
	$ran_{ang}$	points	local	$4^{\circ }$–$8^{\circ }$
Edge information calculation	$edge_{dis}$	points	local	0.04–0.06 (m)
	$edge_{ang}$	points	local	$4^{\circ }$–$8^{\circ }$
	$seg_{dis}$	patches	local and neighborhood	0.06–0.12
	$seg_{ang}$	patches	local and neighborhood	$8^{\circ }$–$18^{\circ }$
Supervoxel segmentation	$super_{dis}$	points, patches	local and neighborhood	0.13–0.2 (m)
	$super_{ang}$	points, patches	local and neighborhood	$13^{\circ }$–$20^{\circ }$
Patch-based region growing	$grow_{dis}$	patches	global	0.2–0.4 (m)
	$grow_{ang}$	patches	global	$15^{\circ }$–$30^{\circ }$

**Table 3 sensors-20-04209-t003:** Execution time for icosahedron.

Method Name	Time (s)
**RG**	2.62
**RHT**	2.31
**PCL-RAN**	2.22
**HT-RG**	32.07
**DSE**	90.81
**MOE**	0.65
**RAN-RG**	1.12
**Ours**	1.31

**Table 4 sensors-20-04209-t004:** Indicators score of Rock1.

Method	Precision	Recall	F1
**RG**	73.1%	68.4%	71.2%
**RHT**	76.3%	78.8%	77.9%
**PCL-RAN**	79.5%	83.3%	81.4%
**HT-RG**	93.2%	89.3%	91.5%
**DSE**	85.4%	79.5%	83.5%
**MOE**	91.9%	91.6%	91.8%
**RAN-RG**	90.5%	91.8%	91.3%
**Proposed Method**	94.9%	97.5%	96.2%

**Table 5 sensors-20-04209-t005:** Indicators score of plane1.

Method	Precision	Recall	F1
**RG**	18.9%	39.7%	25.2%
**RHT**	37.8%	52.2%	43.5%
**PCL-RAN**	29.4%	78.6%	43.8%
**HTRG**	94.5%	74.2%	83.6%
**DSE**	96.5%	66.0%	78.1%
**MOE**	32.3%	56.2%	39.2%
**RAN-RG**	57.6%	45.4%	51.0%
**Proposed Method**	95.1%	87.4%	91.3%

**Table 6 sensors-20-04209-t006:** Execution time for all rocks(s).

Method Name	Rock1	Rock2	Rock3	Rock4
**RG**	12.21	17.09	9.53	16.87
**RHT**	39.97	42.81	48.44	55.38
**PCL-RAN**	28.35	26.17	44.48	47.60
**HTRG**	151.08	162.17	164.78	198.41
**DSE**	302.74	287.15	341.32	388.76
**MOE**	1.13	1.42	2.56	2.75
**RAN-RG**	1.66	2.33	3.50	3.87
**Proposed Method**	4.08	5.27	8.15	9.47

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
