# Peer review of "High-Precision Plane Detection Method for Rock-Mass Point Clouds Based on Supervoxel"

_sensors, 2020, doi:10.3390/s20154209_

Round 1

Reviewer 1 Report

The manuscript demonstrates a plane detection method to construct a 3D rock mass model that can be used in numerical analysis in the field of rock mechanics engineering. In general, the proposed methodology and related references have been thoroughly studied. However, the aim of rock plane detection using photogrammetry method has to be explained to demonstrate the rationale of the study. This can be embedded in Section 1 or Section 2 of the manuscript.  

Regarding Section 3.1.2 and 3.2, there is a reference that used colour and normal vector of the point clouds to distinguish surfaces and edges of rock. Refer, Hyongdoo Jang, Itaru Kitahara, Youhei Kawamura, Yasunori Endo, Erkan Topal, Ryo Degawa & Samson Mazara (2020) Development of 3D rock fragmentation measurement system using photogrammetry, International Journal of Mining, Reclamation and Environment, 34:4, 294-305, DOI: 10.1080/17480930.2019.1585597

Author Response

Thank you for your comments. The content of the article has been modified according to your suggestions. Please check the attachment.

Thank you very much for your help

Reviewer 2 Report

The authors present an approach of rock discontinuities identification from point clouds. The manuscript is clear and easy to understand. However, I am not sure of the novelty of this work. Also, from the view of geotechnical engineering, some more work is still to be added. The details about the reviews of this manuscript are as follows:

  1. The novelty is not enough. Gigli (2011) has proposed a method to semi-automatically extract the rockmass structural data from 3D point clouds. In my opinion, your work are so similar to each other. Comparatively, his work is more practical and professional. Please refer to “Semi-automatic extraction of rockmass structural data from high-resolution LIDAR point clouds”.
  2. In geotechnical engineering or geological engineering, we do not only use the results of plane detection to perform 3D model building or rockmass stability analysis. Normally, several parameters are needed to be determined based on the discontinuity identification for the above-mentioned engineering purposes, such as orientation, spacing, roughness, persistence, aperture. Please refer to “ Suggested methods for the quantitative description of discontinuities in rock masses. International Journal of Rock Mechanics and Mining Sciences & Geomechanics Abstracts1978;15:319–68”.
  3. Please, make a better, broadened review of world literature with reference to rock discontinuities (rock joints) extraction from point clouds, especially in the field of Geotechnical Engineering and Engineering Geology. Please, check the following works (you will find others in scientific dababases):

http://www.sciencedirect.com/science/article/pii/S0098300416306471?via%3Dihub

https://www.sciencedirect.com/science/article/pii/S0013795217317477

http://www.sciencedirect.com/science/article/pii/S0098300416307579?via%3Dihub

http://www.sciencedirect.com/science/article/pii/S0098300415300996

  1. 26-28: “However, in some special fields, the processing precision of previous techniques does not match the precision that can be provided by the data.”-- unclear in the context of the sentence.
  2. 30: In geotechnical engineering or geological engineering, the “planes” are termed as “rock discontinuities”.
  3. 36-37: “The complex surface structures of rock mass bring a great challenge to high-precision plane detection.”----unclear, what is the great challenge?
  4. In the field survey, we also need to know the lithology of the rock outcrop, because the type of rock affects the quality of point clouds (e.g., intensity), which in turn to impact on the parameter selection when performing “plane detection”. Therefore, the geology background about these four cases is suggested to be provided.
  5. Table 7: Running time for all cases is suggested to be listed.

Author Response

Thank you for your comments.

The article has been revised according to your suggestion.

I have answered your questions in the attachment.

What needs to be explained to you is that this work is aimed at the 3D reconstruction of rock mass. I have explained this part in detail in the new manuscript.

In addition, the language of the article has been re edited,which corrected basic language errors without changing the meaning of the original text, and improved the readability of the article.

Wish you success in your work

Reviewer 3 Report

This paper presents a high-precision plane detection approach for 3D rock-mass point clouds, which is claimed to deal with complex surface structures, achieving a high level of details in the plane detection. Plane detection is always an interesting and popular topic in almost any point cloud related applications. This work tried to present a novel supervoxel based method for plane detection, and the proposed idea seems to work well for complicated scenarios (i.e., rock mass) and reveals good results. However, for the current manuscript, there are still some flaws that should be tackled before acceptance.

1. Please use "Supervoxel" instead of "Super-Voxel". "Supervoxel" is already a commonly used terminology.

2. I recommend rewriting the paragraph of the contribution part, highlighting the novelty of your approach and the contribution of using supervoxel structure. Currently, the contribution you mentioned is not convincing. For example, you mentioned that " local coplanarity test based on voxels" is used, however, the "local coplanarity" has already been widely used in many previous voxel-based studies. See the following:
Octree-Based Region Growing for Point Cloud Segmentation. ISPRS Journal of Photogrammetry and Remote Sensing 104: 88–100
Segmentation of building roofs from airborne LiDAR point clouds using robust voxel-based region growing. Remote Sensing Letters 8 (11), 1062-1071
Incremental segmentation of lidar point clouds with an octree‐structured voxel space. The Photogrammetric Record, 26, no. 133 (2011): 32-57.
These studies should be mentioned if you regard the "local coplanarity" as one of the contributions.

3. The literature review part should be restructured as well. I can hardly catch the rules that you used for categorizing the types of methods.
Actually, HT and RANSAC should be categorized in the same type as voting-based methods. The voting of the former one is in the parametric space, while the latter one is in Euclidean space.
Moreover, you should be aware of the difference between region growing based methods and clustering-based methods. Namely, whether growing seeds are needed.
Moreover, some recent work on plane detection from point clouds in complex scenarios are missing:
Plane Segmentation Based on the Optimal-vector-field in LiDAR Point Clouds. IEEE Transactions on Pattern Analysis and Machine Intelligence.
Robust segmentation and localization of structural planes from photogrammetric point clouds in construction sites, Automation in Construction 117, 103206

4. For section 2.4, it is nice to review some work related to rock mass, but simultaneously, please state the importance of rock mass detection in the introduction part.

5. Section 3.2, please check, is it a "clustering"? Or "seed-based region growing"?

6. Table 2. Effective range. What is the difference between "local" and "neighborhood"? Does a neighborhood not include local information?

7. Table 3. There are only abbreviations. What is the meaning of giving such a table?

8. Fig. 5. I cannot observe visual differences.

9. Fig 8. Again, I can hardly see any differences between these results using Icosahedron.

10. Notations and symbols used in the algorithm tables should be checked. There is a rule of describing algorithms with pseudo-codes, but not the way you did.

11. Table 6. Why all the F1 of comparing methods are always too low? Actually comparing results of such a low F1 makes no sense. For region growing, it is quite sensitive to the setting of thresholds. It should not perform as bad as this. Please try to tune the thresholds.

12. English writing of this current manuscript must be improved. The poor English writing makes this manuscript very difficult to read. Thorough proofreading is mandatory. Especially for the use of articles, there are grammar errors of plurals and incorrectly used articles.

Author Response

Thank you for your comments.

Your suggestions are very valuable, especially regarding the concept of clustering and the classification of related work.I have explained your questions in the attachment。

What needs to be explained to you is that this work is aimed at the 3D reconstruction of rock mass. I have explained this part in detail in the new manuscript.

In addition, the language of the article has been re edited,which corrected basic language errors without changing the meaning of the original text, and improved the readability of the article.

I hope you all the best in your work.

Round 2

Reviewer 2 Report

Dear Authors,

I have read the revised version of the manuscript "High-Precision Plane Detection Method for Rock-Mass Point Clouds based on Super-Voxel" submitted for publishing in Sensors Journal. They have addressed the reviewers suggestions, and in the present form the manuscript has been improved. In my opinion, the paper is now suitable to be published.

Best regards.

Reviewer 3 Report

The authors have made significant efforts in the revision of this manuscript. All my concerns have been solved and now I support to accept this work.